# Indirect Effects of the COVID-19 Pandemic on Routine Childhood Vaccination in Low-Income Countries: A Systematic Review to Set the Scope for Future Pandemics

**DOI:** 10.3390/microorganisms12030573

**Published:** 2024-03-13

**Authors:** Jessica E. Beetch, Amanda Janitz, Laura A. Beebe, Mary Gowin, Chao Xu, Shari Clifton, Katrin Gaardbo Kuhn

**Affiliations:** 1Department of Biostatistics and Epidemiology, Hudson College of Public Health, University of Oklahoma Health Sciences, Oklahoma City, OK 73104, USA; jessica-beetch@ouhsc.edu (J.E.B.); amanda-janitz@ouhsc.edu (A.J.); laura-beebe@ouhsc.edu (L.A.B.); chao-xu@ouhsc.edu (C.X.); 2Department of Health Promotion Sciences, Hudson College of Public Health, University of Oklahoma Health Sciences, Oklahoma City, OK 73104, USA; mary-gowin@ouhsc.edu; 3Robert M. Bird Health Sciences Library, University of Oklahoma Health Sciences, Oklahoma City, OK 73104, USA; shari-clifton@ouhsc.edu

**Keywords:** COVID-19, pandemic, vaccination, public health, children, low-income

## Abstract

The COVID-19 pandemic halted progress in global vaccine coverage and disrupted routine childhood vaccination practices worldwide. While there is ample evidence of the vaccination decline experienced during the pandemic, it is less clear how low-income countries were affected. We executed a systematic review to synthesize the current literature on the impacts of routine childhood vaccinations in low-income countries from 1 January 2020 to 8 February 2023. We collected data using an extraction form on Covidence and assessed the quality of studies included in the review using the Risk of Bias in Non-Randomized Studies of Interventions (ROBINS-I) tool. Effect estimates for changes in vaccination during the pandemic were reported and summarized. Factors that influenced changes were grouped into descriptive themes. Thirteen studies, encompassing 18 low-income countries and evaluating 15 vaccines at varying doses, were included in the final review. We found that routine childhood vaccinations during the COVID-19 pandemic varied considerably by vaccine type, location, and phase of the pandemic. Nine different themes were identified as factors that influenced changes in vaccination. Documenting past experiences and lessons learned is crucial for informing preparedness efforts in anticipation of future public health emergencies. Failure to effectively address these things in the next public health emergency could result in a recurrence of declining routine childhood vaccinations.

## 1. Introduction

Global vaccination, particularly in children, has had a profound impact on public health in recent centuries. Routine childhood vaccines have prevented multiple diseases that can be fatal or cause life-long disabilities and are a catalyst for improved overall health. Vaccination programs have successfully reduced the incidence of previously virulent diseases like polio, measles, and influenza and have eradicated smallpox, one of the deadliest diseases in history [1]. Adequate vaccination coverage is essential to create herd immunity. With herd immunity, a sufficient number of individuals are vaccinated so that the disease is less likely to spread to those who cannot be vaccinated, like those who are too young or have certain health conditions. When vaccination coverage drops and herd immunity subsides, outbreaks are more likely to arise [2].

Prior to the onset of the COVID-19 pandemic, vaccination coverage had greatly improved over several decades, reducing the incidence of many vaccine-preventable diseases worldwide [3]. Although vaccines are administered for numerous diseases throughout the world, the third dose of diphtheria, tetanus, and pertussis (DTP 3) vaccine is a good indicator of access to immunization services and is therefore commonly used as a metric for global vaccination coverage [4]. DTP 3 global vaccination coverage remained at 86% from 2016 to 2019 after steadily increasing over recent decades. However, global vaccination coverage for DTP 3 decreased to 83% in 2020 (ranging from 72% in Africa to 95% in the Western Pacific) [5] and 81% in 2021 (ranging from 71% in Africa to 94% in Europe) [6]. Even a small drop in global vaccination coverage implies that millions of children did not receive the DTP 3 vaccine and other important routine vaccinations. With the emergence of COVID-19 came a shift in prioritization of services, resulting in delays in medical care and declines in vaccination coverage [7]. Although vaccine services were delayed for all ages, including adults, the largest declines in vaccination were observed in children [8]. The United Nations International Children’s Emergency Fund (UNICEF) described a major backslide in childhood vaccinations caused by the pandemic [9]. In 2020, 23 million children missed routine vaccinations, almost four million more than in 2019. A majority of those children did not receive a single vaccine of any type, widening vaccine inequities. This worsened in 2021, when global vaccine coverage was the lowest it had been since 2007 [10]. Even in countries that had widespread access to the COVID-19 vaccine, other routine vaccinations slipped, leaving children at risk for preventable diseases [9]. A multitude of factors contribute to reductions in vaccination coverage, including local vaccination culture, vaccine hesitancy, and vaccine mandates, as well as access to care and adequate vaccine supply [11]. Vaccination coverage started to recover in 2022 when DTP 3 vaccine coverage increased to 84% globally (Africa = 72%, Americas = 83%, Eastern Mediterranean = 84%, Europe = 94%, South East Asia = 91%, and Western Pacific = 93%) [12]. However, millions of children were still missing out on vaccines compared to before the pandemic began. Low-income countries experienced a slower recovery, with some areas encountering ongoing declines and stagnant trends. Recovery has not been equal, even within low-income countries, due to inequitable jobs and welfare and different levels of urgency to ensure vaccine accessibility after the COVID-19 lockdowns. The speed of recovery has also been influenced by high populations of children and civil unrest within countries [13]. The pandemic has been a stark reminder that vaccine distribution has been inequitable throughout time, and the poorest children continue to be the least likely to receive vaccines [4].

While it is known that the COVID-19 pandemic impacted routine childhood vaccination, with declines in vaccine administration seen worldwide [14], these effects have not been thoroughly studied in all major population segments. Research is often heavily concentrated in high-income countries, with a lack of attention to outcomes in lower-income countries [15]. Previous reviews have identified disruptions to routine vaccination practices globally [16,17] or in both low- and middle-income countries [18,19] but have never specifically focused on the impacts to low-income countries. Although routine vaccination disruption in low-income countries has been discussed as a concern [20], the topic has not been synthesized in a systematic review. Further, no papers in the literature have studied the quantitative impacts of the pandemic on routine vaccination combined with the qualitative suspected factors influencing changes in vaccination practices in low-income countries. It is essential to study populations that are vulnerable to poverty, such as low-income countries, to gather information about their unique situations and the contributing factors to vaccine uptake. Populations vulnerable to poverty often face disproportionate outcomes in mental health, birth defects, and chronic and infectious diseases and experience systemic inequalities, like discrimination and differential resources, which increase their risk of severe health outcomes [21]. Although childhood poverty persists throughout the world, even in high-income countries, it is most prevalent in lower-income countries located throughout Sub-Saharan Africa and South Asia [22]. The pandemic has emphasized the necessity to address health inequities and conduct research on these low-income groups.

Consequences emerge when routine vaccination practices change, especially in low-income areas. When vaccination declines, there is a higher risk of contracting vaccine-preventable diseases and the potential for outbreaks increases. These adverse outcomes, along with other social and economic factors such as reduced education, employment, and safety in low-income populations, exacerbate their health risk. It is not only important to understand the extent to which the pandemic impacted routine childhood vaccinations, but also which populations bore the impacts disproportionately. To the best of our knowledge, this systematic review will be the first to address the pandemic’s impact on routine childhood vaccinations in low-income countries worldwide by collecting relevant published evidence on this topic. The overall goal of this review is to increase understanding of how children in low-income countries worldwide experienced indirect impacts on their health due to the COVID-19 pandemic. It is vital to study the pandemic’s impacts and identify high-risk groups to prepare for the next large outbreak or public health emergency.

The objectives of this systematic review were to identify and evaluate how the COVID-19 pandemic impacted routine childhood vaccinations in low-income countries by (i) identifying changes in vaccination rates and (ii) identifying reported factors that led to changes in vaccination rates in studies published from 1 January 2020 to 8 February 2023.

## 2. Materials and Methods

### 2.1. Protocol

We prepared the systematic review protocol using guidance from Preferred Reporting Items for Systematic Reviews and Meta-Analyses (PRISMA 2020) [23] with additional recommendations from Conducting Systematic Reviews and Meta-Analyses of Observational Studies of Etiology (COSMOS-E) [24]. We registered the protocol with the International Prospective Register of Ongoing Systematic Reviews, United Kingdom (PROSPERO CRD42023491742).

### 2.2. Eligibility Criteria

Studies were selected according to the following criteria:

#### 2.2.1. Population

This review focused on the impacts of the pandemic on children in low-income countries. It only included studies focusing on persons 18 years of age and younger residing in countries with a low-income economy (USD $1085 or less as of February 2023). The 28 low-income countries worldwide include Afghanistan, Burkina Faso, Burundi, the Central African Republic, Chad, the Democratic Republic of the Congo, Eritrea, Ethiopia, Gambia, Guinea, Guinea-Bissau, Liberia, Madagascar, Malawi, Mali, Mozambique, Niger, North Korea, Rwanda, Sierra Leone, Somalia, South Sudan, Sudan, Syrian Arab Republic, Togo, Uganda, Yemen, and Zambia [25]. All genders and racial/ethnic groups were eligible for inclusion.

#### 2.2.2. Exposure and Comparison

We included studies published from 1 January 2020, the month after the first COVID-19 case was discovered, until 8 February 2023, the day before the search took place. The exposure of interest was the three-year time period (1 January 2020 to 8 February 2023) that included COVID-19 emergence and a significant portion of the COVID-19 pandemic. The comparison time period (31 December 2019 and before) included the time prior to COVID-19 emergence. All included studies were needed to compare outcomes during the COVID-19 time period to those of the pre-COVID-19 time period.

#### 2.2.3. Outcome

The outcomes of interest were two separate components related to routine childhood vaccination practices. These components included potential quantitative changes in vaccination rates and qualitative factors that led to the changes. The definition and calculation of changes in routine childhood vaccination vary between locations. Therefore, for this review, we considered all routine vaccinations that were reported as relevant for each study population and location.

#### 2.2.4. Study Design

We included retrospective cohort studies, case–control studies, ecological studies, cross-sectional studies, and papers analyzing local or national surveillance data.

#### 2.2.5. Language

We only included studies that were reported in the English language.

#### 2.2.6. Exclusions

We excluded studies not covering routine childhood vaccinations, papers not published in English, studies that reported results from only high- or middle-income countries, and studies that did not directly relate the COVID-19 pandemic to the indirect impacts on childhood vaccinations. Further, we excluded systematic and narrative reviews, modeling studies, single case studies, laboratory studies, letters, reports, editorials, commentaries, and studies with only qualitative results, such as surveys with caregiver opinions.

### 2.3. Search Strategy

We searched the available literature on Ovid MEDLINE^®^ and Epub Ahead of Print, In-Process, In-Data-Review, and Other Non-Indexed Citations. The search was executed on 9 February 2023. To focus on impacts of the COVID-19 pandemic, the search was limited to studies published from 1 January 2020 to 8 February 2023. The search used a combination of controlled vocabulary terms and keywords to incorporate the following concepts of “immunization, vaccination or vaccine, low-income, SARS-CoV-2, COVID-19, pediatric or child or infant or neonate or adolescent or teen or youth, and routine or schedule or catchup”, the search functions “exploded and multi-purpose”, and Boolean operators “and, or”. Search strategy and keywords were discussed and finalized with the Associate Director and Head of Reference and Instructional Services at the Robert M. Bird Health Sciences Library (S.C.).

### 2.4. Result Screening and Selection

Screening and selection of articles were conducted using the web-based application Covidence [26]. Articles produced by the literature search were screened for inclusion using a two-step process. Firstly, articles were screened based on their title and abstract to remove duplicates and irrelevant papers. Secondly, the remaining papers required full-text review to determine eligibility. Articles were excluded if they did not meet the eligibility criteria, including those with ineligible study designs, outcomes, and/or populations. Screening and selection of articles were completed independently by two reviewers to reduce error. Discrepancies were discussed and resolved by the two-reviewer team (J.E.B. and K.G.K.).

### 2.5. Data Collection

Data collection was completed using a customized extraction form developed on Covidence [26]. Data collection questions were developed using data extraction guidelines from COSMOS-E. This included the collection of bibliographic information, study design, study participant characteristics, exposure(s) and outcomes, and effect measures from each included paper [24]. Data items were collected independently by one reviewer (J.E.B.) and then examined for errors by a second reviewer (K.G.K.).

### 2.6. Data Items

Using the customized data collection form, the data extracted from each paper are itemized in Table 1.

### 2.7. Risk of Bias Assessment

We appraised the methodological quality of the papers included in the review using bias domains and scoring categories from the Risk of Bias in Non-Randomized Studies of Interventions (ROBINS-I) tool by Cochrane [27]. ROBINS-I bias domains used in the assessment included bias due to confounding, bias in the selection of participants into the study, bias in the measurement of exposures and outcomes, bias due to missing data, and bias in the selection of studies or reported results. Scoring categories for risk of bias judgement for each domain were low risk, moderate risk, and high risk. The risk of bias assessment was conducted independently by two reviewers. Discrepancies were discussed and resolved by the two-reviewer team (J.E.B. and K.G.K.).

### 2.8. Data Synthesis

We anticipated considerable differences in study periods, vaccine types, and analyses between studies; therefore, a meta-analysis was deemed inappropriate due to substantial heterogeneity. However, effect estimates for quantitative changes in vaccination during the pandemic were reported and summarized by ‘decline in vaccination’ and ‘increase in vaccination’ groups. We also conducted a narrative synthesis of factors that influenced changes in effect estimates. We extracted these factors from results, findings, or discussion sections from each study included in the review. We entered the factors extracted from the studies verbatim into the Covidence extraction form. Finally, we manually grouped the reoccurring factors and developed descriptive themes [28].

## 3. Results

### 3.1. Study Selection

We identified 118 studies from the extended MEDLINE file from 1 January 2020 to 8 February 2023 (Appendix A). After removing one duplicate paper, we excluded 68 papers during title and abstract screening that did not meet eligibility criteria for reasons such as ineligible outcomes, population, location, and study design (Figure 1). Four of the 49 remaining papers were unable to be accessed or retrieved. We completed a full-text review on 45 papers where studies were excluded due to outcomes not pertaining to routine vaccination practices (n = 6), studies focusing on adult populations instead of children (n = 3), wrong study design (n = 22), and duplicate outcomes due to use of the same vaccine data (n = 1). The final review included 13 studies in total (Figure 1).

### 3.2. Study Characteristics

The 13 studies selected for inclusion assessed changes in routine vaccination in 18 low-income countries including Afghanistan, Burundi, the Central African Republic, Chad, the Democratic Republic of the Congo, Eritrea, Ethiopia, Gambia, Liberia, Malawi, Mali, Mozambique, Rwanda, Sierra Leone, Somalia, South Sudan, Uganda, and Yemen (highlighted in dark blue in Figure 2). Some studies were set in capital cities, while others examined part of a country, a single country, or a group of countries. While all 13 studies assessed routine vaccination changes in children, one paper specifically studied newborns [29], one paper studied children younger than one year of age [30], and two papers studied children younger than five years of age [31,32]. The studies assessed 15 different vaccines at varying doses including Bacillus Calmette–Guerin for tuberculosis (BCG), polio (Pol), inactivated polio vaccine (IPV), oral polio vaccine (OPV), combined diphtheria, tetanus, and pertussis, hepatitis B, Haemophilus influenzae type b (DTP Hep B Hib), diphtheria, tetanus, and pertussis (DTP), hepatitis B (Hep B), pneumococcal, pneumococcal conjugate vaccine (PCV), rotavirus (Rota), measles-containing vaccine (MCV), measles and rubella, pentavalent (Penta), malaria intermittent preventive treatment (IPTi) and yellow fever, while some studies examined overall vaccination or fully vaccinated status. The study periods ranged from January 2016 to February 2021 but were published between September 2020 and December 2022. Methodologically, five studies used a cross-sectional study design [29,31,32,33,34], five used an interrupted time series study design [35,36,37,38,39], two used a cohort study design [30,40], and one used a mixed methods study design [41].

### 3.3. Changes in Vaccination

The number of children who received routine vaccinations during the COVID-19 pandemic varied considerably by vaccine type, location, and phase of the pandemic (Table 2). Although declines in vaccination practices were seen at some point in the pandemic in every low-income country studied, some countries experienced nonsignificant changes while others observed increases in vaccination during the pandemic compared to before it began. The effect estimates captured include measures of mean values, observed values, and percent change (Table 3).

#### 3.3.1. Declines in Vaccination

Laghman, Afghanistan, faced a 21.4% (*p* < 0.001) decline in overall vaccination from April to July 2020 [31]. Liberia experienced a 17.0% decline in overall vaccination coverage (95% CI: −39.1, −8.0) early in the pandemic from March to August 2020, while Malawi experienced a 9.0% decline in overall vaccination coverage (95% CI: −18.0, −3.9) later in the pandemic from September 2020 to February 2021 [30]. Other settings remained stable in children fully vaccinated by 1 year of age throughout Ethiopia (−0.9, 95% CI: −3.9, 2.1) [39], children fully vaccinated in Addis Ababa, Ethiopia (−0.6, *p* = 0.95) [34], and overall vaccination coverage (−20.0%, *p* = 0.197) and complete vaccination (−18.0%, *p* = 0.544) in Nampula, Mozambique [41]. Vaccine clinic attendance declined from pre-pandemic baseline levels in Kampala, Uganda, with almost 5000 fewer attendees during the pandemic (*p* = 0.04) [38].

Laghman, Afghanistan, reported 6.0% to 28.0% (*p* < 0.001) declines in BCG, DTP 2–3, Hep B, IPV, MCV, OPV 0–4, PCV 1–3, Penta 1, and Rota 1–2 vaccinations from April to July 2020 [31]. West Rural Area, Sierra Leone, experienced large declines in vaccinations for all vaccines studied, ranging from a 51.1% to 83.7% (*p* < 0.0005) drop in BCG, OPV 0–3, Penta 1–3, PCV 1–3, Rota 1–2, IPTI 1–3, IPV, MCV 1–2, and yellow fever vaccinations in March and April 2020 [32]. The third dose of the Penta vaccine declined in Yemen in February, April, May, and June of 2020, with the strongest declines seen in May 2020 (−24.5, 95% CI: −30.6, −18.4) and June 2020 (−15.3, 95% CI: −20.2, −10.5) [40]. Vaccinations also declined for the third dose of the Penta vaccine in Liberia (−7.8%, 95% CI: −13.0, −2.5), Mali (−17.4%, 95% CI: −22.6, −12.3), and Sierra Leone (−12.6%, 95% CI: −19.1, −6.1) and for BCG in Mali (−11.8%, 95% CI: −15.4, −8.2) and Sierra Leone (−7.4%, 95% CI: −11.9, −2.9) from March to July 2020 [35]. Utilization rates declined in Rwanda for BCG, Pol 0–2, DTP Hep B Hib 1–2, Pneumococcal 1–2, and Rota 1–2 vaccines (*p* = 0.001 to *p* = 0.009) in March and April 2020 [33]. Vaccinations remained stable for Penta 3 and BCG vaccines in the Democratic Republic of the Congo (−0.1%, 95% CI: −1.2, 1.0, and −1.4%, 95% CI: −3.4, 0.6, respectively) and Somalia (−3.6%, 95% CI: −9.8, 2.6, and −2.4%, 95% CI: −8.9, 4.2, respectively) from March to July 2020 [35]. Declines were seen in the third dose of the DPT vaccine in the Central African Republic (−3.0%), South Sudan (−7.0%), Burundi (−12.0%), Eritrea (−9.0%), and Rwanda (−2.0%) and in the first dose of MCV in the Central African Republic (−3.0%), Burundi (−20.0%), and Rwanda (−4.0%) from April to June 2020 [37]. Further, rural Gambia experienced declines in vaccination during the interruption period (April to June 2020), the initial recovery period (July to September 2020), and the late recovery period (October 2020 to December 2020) [36]. Declines were reported in vaccine administration (interruption: −38.3%, initial recovery: −15.1%) and for BCG (interruption: −47.2%, initial recovery: −20.0%), Hep B (interruption: −46.9%, initial recovery: −20.0%), Penta 1 (interruption: −43.1%, initial recovery: −33.0%), OPV 1 (interruption: −83.6%, initial recovery: −34.0%), PCV 1 (interruption: −42.4%, initial recovery: −33.0%, late recovery: −2.0%), and Rota 1 (interruption: −43.4%, initial recovery: −34.0%, late recovery: −2.6%) vaccinations [36].

#### 3.3.2. Increases in Vaccination

While many low-income countries reported declines in routine childhood vaccination during the COVID-19 pandemic, some experienced increases in routine vaccination at differing phases throughout the pandemic. Overall vaccination coverage increased in Malawi early in the pandemic (13.7%, 95% CI: 2.4, 33.6) from March 2020 to August 2020 compared to before the pandemic began. Overall vaccination coverage was stable in Liberia late in the pandemic (27.0%, 95% CI: −12.5, 56.5) from September 2020 to February 2021 and throughout the whole pandemic study period (8.0%, 95% CI: −12.1, 15.9) [30].

In Malawi, BCG vaccination increased (5.6%, 95% CI: 0.5, 10.8), but Penta 3 vaccination was stable (1.2%, 95% CI: −2.5, 4.9) alongside BCG vaccination in Liberia (0.3%, 95% CI: −5.2, 5.8) from March to July 2020 [35]. Although declines in vaccination were experienced in other months of the pandemic in Yemen, Penta vaccinations were stable in March 2020 (0.02, 95% CI: −4.3, 4.4) [40]. In Rwanda, the vaccination utilization rate of the measles + rubella vaccine remained stable in March and April 2020 (0.014, *p* = 0.642) [33], while in Addis Ababa, Ethiopia, MCV vaccination coverage change was also stable from April to June 2020 (1.7, *p* = 0.86) [34]. Additionally, the first dose of MCV increased in Chad (13.0%), the Democratic Republic of the Congo (2.0%), and Eritrea (2.0%), and the third dose of DPT increased in Chad (6.0%) and the Democratic Republic of the Congo (1.0%) from April to June 2020 [37]. Finally, in rural Gambia, increases were observed during the initial recovery period from July to September 2020 for MCV (79.0%) and yellow fever (88.9%) vaccinations. During the late recovery period from October to December 2020, vaccine administration (1.9%), BCG (3.0%), Hep B (2.5%), and OPV 1 (22.7%) vaccination increased [36].

### 3.4. Factors That Influenced Changes in Vaccination Rates

We summarized the proposed factors that influenced changes in vaccination rates in low-income countries during the pandemic and grouped them into nine different themes, as shown in Table 4.

#### 3.4.1. Communication Challenges

In Rwanda, reduced communications on routine health services led to declines in vaccinations [33]. Conversely, increased high-risk communication during COVID-19 in Laghman, Afghanistan, made people reluctant to leave their homes and seek routine health services, leading to a decline in vaccinations [31]. The Democratic Republic of the Congo, Liberia, Malawi, Mali, Sierra Leone, and Somalia experienced false social media claims, such as voice notes on WhatsApp warning mothers not to visit immunization clinics and vaccinate their babies with Western-developed COVID-19 vaccines. Clinic attendance later began to improve with radio programs and community visits to dispel rumors [35].

#### 3.4.2. Fear of COVID-19

Fear of COVID-19 was reported as a widespread issue during the pandemic across multiple low-income countries. Many people were hesitant to visit health facilities, hospitals, or immunization clinics due to fear of infection or transmission of the virus. Fear of contagion was suggested as a reason for vaccination declines in parts of Ethiopia [34,39], Sierra Leone [32], Uganda [38], Liberia and Malawi [30], Yemen [40], Gambia [36], and Rwanda [33].

#### 3.4.3. Financial Barriers

Financial barriers to seeking routine health services were proposed as a factor contributing to declines in vaccination during the pandemic in Addis Ababa, Ethiopia [34]. Throughout Ethiopia, many were unable to pay for health services due to loss of employment or compensation [39].

#### 3.4.4. Health System Practices

The differences in vaccination rates experienced between low-income countries were, in part, from differences in the resilience and structure of their health systems [35]. In the Central African Republic, Chad, the Democratic Republic of the Congo, South Sudan, Burundi, Eritrea, and Rwanda, the countries that had weaker health systems were more vulnerable to disruptions. Countries with previously high vaccination coverage did a better job maintaining those levels, but those with previously lower coverage saw larger declines [37]. In South West Ethiopia, there were inadequate newborn care practices [29], whereas, in Nampula, Mozambique, wait times for care increased during the pandemic [41]. An obstacle widely experienced was the intentional suspension of routine care to allow room for patients with COVID-19 [32]. However, in some places like the Rural Western Area, Sierra Leone, health systems maintained the same activities as before the pandemic began [32].

#### 3.4.5. Lockdowns and Emergency Measures

Strict or lengthy lockdowns caused barriers to receiving routine health services and greatly influenced changes in vaccination during the pandemic, as reported in Ethiopia [39], the Central African Republic, Chad, the Democratic Republic of the Congo, South Sudan, Burundi, Eritrea, and Rwanda [37]. In rural Gambia, there were no lockdowns, but the state of emergency raised alarms and led to restricted movement [36]. In Laghman, Afghanistan, border closures also restricted movement to inhibit the attendance of routine health services [31]. Restrictions differed by district in Liberia and Malawi but were not strictly adhered to in rural districts [30]. Yemen did not enforce lockdowns or restrict movement [40].

#### 3.4.6. Pre-Existing Factors

Other contributing factors that influenced vaccination were compounded by pre-existing challenges in low-income countries. Pre-existing poor-quality services, poor road conditions, and lack of infrastructure for the production of medical supplies were reported as challenges in Rwanda [33]. The Democratic Republic of the Congo, Liberia, Malawi, Mali, Sierra Leone, and Somalia differed in outcomes because of their prior experiences with epidemics and restrictions [35].

#### 3.4.7. Supply Shortages

Global suspension of vaccine companies and altered supply chains led to shortages in stock of vaccines and other supplies [31]. Addis Ababa, Ethiopia, experienced inadequate supplies of PPE [34], while Liberia and Malawi saw declines in vaccination due to shortages of vaccine stock [30]. The first dose of OPV was out of stock nationally from April to July 2020 in rural Gambia, causing major declines in the administration of the vaccine [36]. The Rural Western Area, Sierra Leone, reported experiencing no problems regarding vaccine supply [32].

#### 3.4.8. Transportation Barriers

Travel restrictions and restricted movement were reported as factors influencing vaccination during the pandemic in Liberia and Malawi [30], Rwanda [33], and rural Gambia [36]. Kampala, Uganda, experienced closures of public transport and increased transport prices [38]. Additionally, a decrease in the use of motorcycle ambulances was reported in Nampula, Mozambique [41].

#### 3.4.9. Workforce Changes

A redirected workforce toward COVID-19 care was experienced in Addis Ababa, Ethiopia [34]. The overall number of health professionals decreased in Nampula, Mozambique [41], while a lack of healthcare staff was reported in Kampala, Uganda [38].

### 3.5. Risk of Bias in Individual Studies

Using bias domains and scoring categories from the ROBINS-I tool, all studies had low or moderate risk of bias and were included in the synthesis of data. Most studies (92%) were categorized as having a moderate risk of bias due to missing data. However, a majority of the studies were categorized as low risk of bias due to confounding (92%), bias in the selection of participants into the study (92%), bias in the measurement of exposures or outcomes (85%), and bias in the selection of studies or reported outcomes (92%). We included a summary table of the risk of bias assessment in Appendix A.

## 4. Discussion

This comprehensive systematic review of 13 studies summarized the available data on changes in routine childhood vaccination in low-income countries during the COVID-19 pandemic. The review bridges gaps in the literature by addressing specific pandemic-driven impacts on children and summarizing the factors that influenced those impacts in low-income countries with residents that are vulnerable to poverty. Changes in vaccination practices, like vaccine coverage and clinic attendance, and influential factors were systematically examined using PRISMA 2020 guidelines and COSMOS-E recommendations. Furthermore, included studies were assessed for bias using a ROBINS-I tool that covered relevant domains of bias for our review. While vaccine types, vaccination changes, and influential factors varied across included studies, our results provide an account of how children in low-income countries weathered the pandemic. The results of this review reveal that most low-income countries experienced a reduction in routine childhood vaccination early in the COVID-19 pandemic. Several factors were reported as proposed influences on routine vaccination in children during the pandemic. Notable factors include fear of contracting COVID-19 in health facilities, changes in health system practices, and transportation barriers.

Declines in vaccination practices were reported for a majority of vaccines in most low-income countries studied. While some declines in vaccination were small or did not reach statistical significance, some declines were extreme. Sierra Leone and Afghanistan saw the largest declines in routine vaccination during the pandemic. The West Rural Area, Sierra Leone, experienced a substantial 84% decline in MCV 2 vaccination and over 50% decline in nine other vaccinations in March and April 2020. Laghman, Afghanistan, experienced large declines in overall vaccinations and nine individual vaccinations from April to July 2020. Declines were observed at least once for all individual vaccines studied and for overall vaccination coverage, fully vaccinated status, and vaccine clinic attendance. Increases reported during later months of the pandemic provide evidence of effective catch-up routines implemented to counteract downward trends. Given the extensive declines in vaccination rates observed in low-income countries during the pandemic, robust catch-up routines were imperative to restore progress from previous vaccination efforts.

Most of the included studies focused on the initial impact period of the pandemic and assessed the first few months after the pandemic was declared. All of the studies began their pandemic study period in March or April 2020, and most ended the study period by July 2020. Some studies chose not to include March 2020 in their study, while others included the month in the baseline study period. Most of the studies that did include March 2020 in the pandemic study period did not report results by month but instead reported totals for the entire study period. COVID-19 was declared a global pandemic by the World Health Organization (WHO) on March 11, 2020, and there was likely a delay in adverse outcomes, prompting studies to handle pandemic start times differently. Studies that divided the pandemic into phases found declines and increases in vaccination depending on the phase of the pandemic.

Changes in routine childhood vaccination practices are not a direct effect of the pandemic but are indirect results due to several influential factors. Low-income countries face distinctive factors influencing routine healthcare and vaccination. These populations often grapple with limited resources, inadequate healthcare infrastructure, and a higher prevalence of disease. Additionally, socioeconomic disparities contribute to challenges in delivering routine healthcare. Outside of the COVID-19 pandemic, factors that generally contribute to vaccine coverage inequality in low- and middle-income countries include maternal and paternal education, household wealth, ethnicity, caste, and area of residence. Urban–rural differences and distance to care influence the number of children who are vaccinated in these populations [19]. The COVID-19 pandemic revealed numerous factors that were reported as contributors to changes in routine vaccination practices during the pandemic in low-income countries. Routine childhood vaccination declines are more likely when there are more influential factors acting against it.

Vaccination rates during this period may have also been influenced by factors unrelated to COVID-19. Civil unrest and military activities from late 2019 to early 2023 likely impacted the well-being and health of people in certain low-income countries during this time. From 2020 to 2022, Ethiopia fought in the Tigray War between the Ethiopian government and the Tigray People’s Liberation Front. It was one of the deadliest conflicts in recent history and set a record for the most people displaced in any country in a single year (2021) [42]. In May 2021, the United States began withdrawing from Afghanistan after two decades of military presence. With this came a subsequent change in government in Afghanistan. The Taliban took control of the capital city in August 2021, raising concerns about the rights and safety of Afghan citizens [43]. In addition to these conflicts, there was ongoing instability in the Democratic Republic of the Congo, Somalia, South Sudan, and Yemen.

There are some limitations that merit consideration when interpreting the results of this review. The primary limitation was the heterogeneity of vaccine types and analyses presented. A meta-analysis was considered but deemed inappropriate since we were unable to pool the effect estimates that were extracted. While we did not conduct a meta-analysis, we compiled and summarized the trends of both declines and increases in vaccination. Secondly, we did not present outcomes from all 28 low-income countries throughout the world due to a lack of available literature. We did include studies from 18 low-income countries that covered WHO African and Eastern Mediterranean Regions. The only region that included a low-income country that we were unable to obtain data from due to lack of papers is the South East Asian Region, which contains North Korea. We did not identify any drastic differences in social and economic facets and healthcare systems between the 18 countries included in this review and the 10 countries that were not included in this review apart from the distinct social and economic structures in North Korea. The countries we missed could have impacted our results, and the review may not be fully representative of what occurred in all low-income countries during the pandemic. Thirdly, there is the potential for selection bias since our study only included papers written in English. Because we studied routine childhood vaccinations throughout the world, it is possible that we failed to include literature written in languages other than English. However, a substantial portion of scientific literature is either in English or has been translated. Although not including studies that use other languages could introduce bias, only using English language studies allowed us to remain consistent throughout the review process. There is an additional potential source of selection bias that may have influenced our results, stemming from the process by which the identified papers were screened and selected. We significantly reduced the risk of bias by involving two reviewers in the screening and selection process. Using the ROBINS-I tool for risk of bias assessment, we found that a majority of studies included in the review had a low risk of bias in confounding, selection bias and measurement domains, suggesting the study findings are reliable. In several studies, we found a moderate risk of bias due to missing data collected from health facilities. No studies featured a high risk of bias for any of the bias domains, and all studies were included in the synthesis. Despite these limitations, we believe that this review successfully summarizes the current literature on routine childhood vaccination during the COVID-19 pandemic and identifies unique factors for changes in routine vaccination in low-income countries.

Although the next public health emergency may differ from COVID-19, we can use the lessons learned from this pandemic to guide practices for children in low-income settings. Compiling health outcomes during the pandemic and the reasons behind those outcomes helps inform preparedness efforts targeted at children who live in settings that put them at an increased risk of becoming infected or being exposed to poor outcomes. The COVID-19 pandemic has taught us to maintain strong communication with the community and try to mitigate fears, ensure transportation alternatives, and keep adequate stockpiles of essential supplies. It has also underscored the importance of maintaining routine health services such as vaccination and avoiding a complete shift of workers to emergency care, as it may result in the breakdown of other essential areas. If these factors are not effectively addressed during the next pandemic or public health emergency, we will likely again see declines in routine childhood vaccination, resulting in illness and mortality from preventable diseases.

## 5. Conclusions

Changes in routine childhood vaccination practices during the COVID-19 pandemic varied among low-income countries, but most saw declining vaccination trends, particularly during the initial months of the pandemic. Changes in routine childhood vaccination practices are indirect consequences due to multiple clinical, infrastructural, and social factors. Primary influential factors reported in low-income countries consisted of fear of contracting COVID-19 in health facilities, changes in health system practices like redirected care to patients with COVID-19, and transportation barriers. Although the COVID-19 public health emergency has been declared over, cataloging previous outcomes and lessons learned will help inform preparedness efforts for individuals vulnerable to poverty in the event of the next pandemic or public health emergency.

## Figures and Tables

**Figure 1 microorganisms-12-00573-f001:**
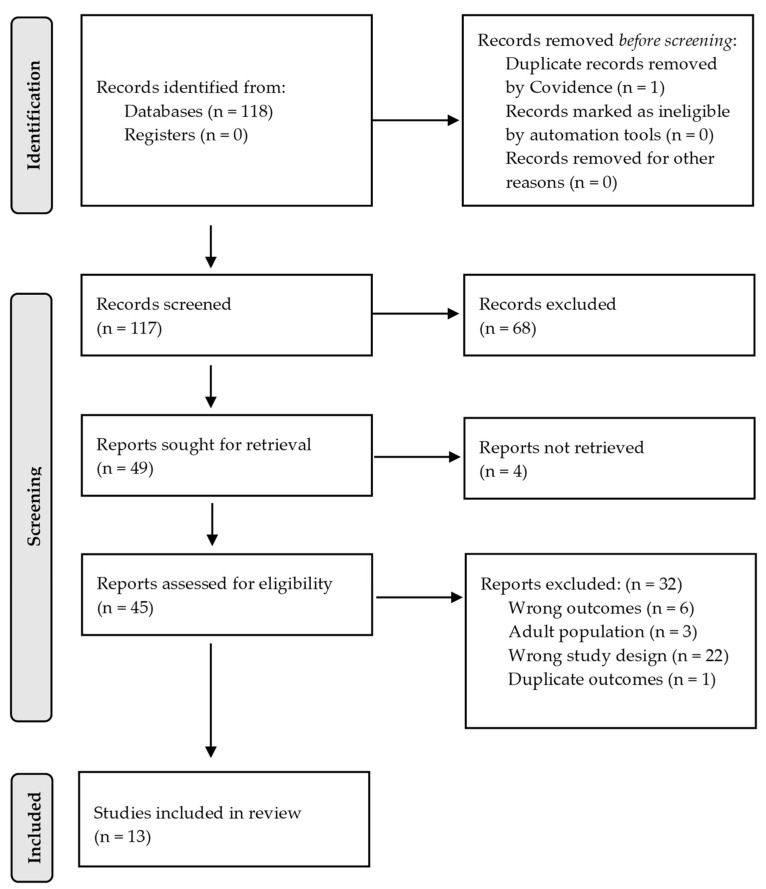
PRISMA 2020 flow diagram of study selection.

**Figure 2 microorganisms-12-00573-f002:**
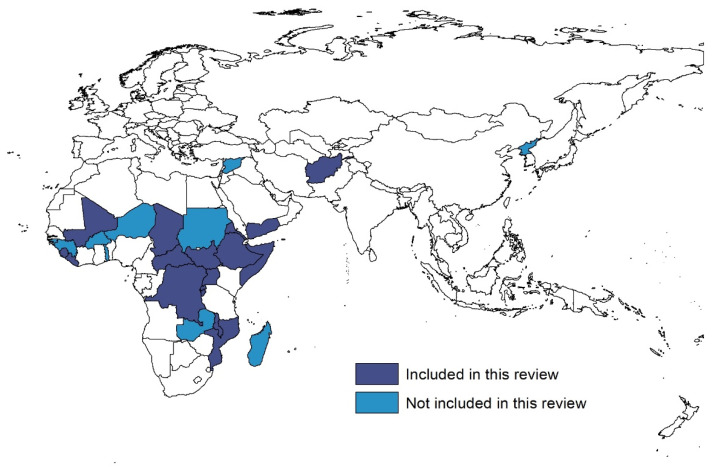
The 28 low-income countries worldwide, as defined by the World Bank.

**Table 1 microorganisms-12-00573-t001:** Data items collected from included papers.

Type	Item
Bibliographic information	Authors
	Publication month and year
	Digital object identifier (DOI)
Study design	Assessment of the study design that was completed
Study participant characteristics	Geographic location of study
	Age groups studied
	Population description
Exposure(s) and outcomes	Time period of the study
	Vaccine(s) studied
Effect measures	Type of measure
	Effect estimate
	Measure before the COVID-19 pandemic, if available
	Measure during the COVID-19 pandemic, if available
Qualitative factors	Factors associated with changes in measurement

**Table 2 microorganisms-12-00573-t002:** Study characteristics of included studies, (n = 13).

Citation	Location	Time Period	Age Group	Vaccines	Key Findings *	
Abid 2022 [31]	Laghman, Afghanistan	April–July 2019 (baseline) compared to April–July 2020 (pandemic)	Children under 5	Overall vaccination, BCG, DTP, Hep B, IPV, MCV, OPV, PCV, Penta 1, Rota 1–2	Significant declines in vaccination for all vaccines studied	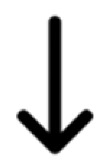
Arsenault 2022 [39]	Ethiopia	January 2019–March 2020 (baseline) compared to April–December 2020 (pandemic)	Children	BCG, Penta, Pneumococcal, Rota, MCV	Non-significant declines in vaccination for all vaccines studied	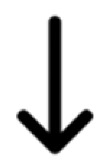
Buonsenso 2020 [32]	Rural Western Area, Sierra Leone	1 March–26 April 2019 (baseline) compared to 1 March–26 April 2020 (pandemic)	Children under 5	BCG, OPV 0–2, Penta 1–3, PCV 1–3, Rota 1–2, IPTi 1–3, IPV, MCV 1–2, Yellow fever	Significant declines in vaccination for all vaccines studied	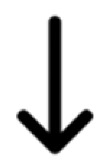
Burt 2021 [38]	Kampala, Uganda	July 2019–March 2020 (baseline), April–June 2020 (pandemic)	Children	Overall vaccination	Significant decline in immunization clinic attendance	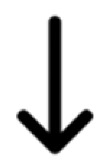
Connolly 2022 [30]	Liberia and Malawi	January 2016–February 2020 (baseline) compared to March 2020–August 2020 (early pandemic) and September 2020-February 2021 (late pandemic)	Children under 1	BCG, OPV or IPV 0–3, Penta 1–3, PCV 1–3, Rota 1–2, MCV	Non-significant decline in vaccination in Malawi, and non-significant increase in vaccination in Liberia	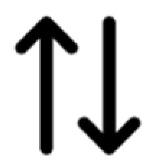
das Neves Martins Pires 2021 [41]	Nampula, Mozambique	March–May 2019 (baseline) compared to March–May 2020 (pandemic)	Children	Overall vaccination	Non-significant declines in vaccination	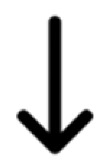
Gebreegziabher 2022 [34]	Addis Ababa, Ethiopia	July–September 2019 (baseline) compared to April–June 2020 (pandemic)	Children	Penta 1, Penta 3, MCV, fully vaccinated	Non-significant decline in Penta 1 and 3 vaccination and fully vaccinated status, and non-significant increase in MCV vaccination	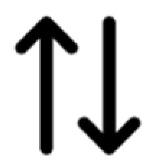
Kassie 2021 [29]	South West Ethiopia	March–June 2019 (baseline) compared to March–June 2020 (pandemic)	Newborns	Penta 1, MCV 1	Significant declines in vaccination for both vaccines studied	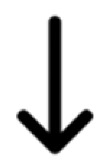
Kotiso 2022 [40]	Yemen	January–June 2019 (baseline) compared to February–June 2020 (pandemic)	Children	Penta 3	Significant decline in vaccination in February, April, May and June, and non-significant increase in March	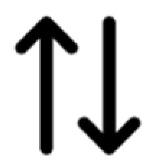
Masresha 2020 [37]	Central African Republic, Chad, Democratic Republic of the Congo, South Sudan, Burundi, Eritrea, Rwanda	January–March 2020 (baseline) compared to April–June 2020 (pandemic)	Children	DPT 3, MCV 1	Declines and increases in vaccination for both vaccines	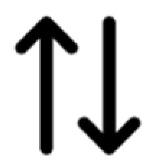
Osei 2022 [36]	Rural Gambia	1 September 2019–31 March 2020 (baseline), 1 April–30 June 2020 (interruption), 1 July–30 September 2020 (initial recovery), 1 October–31 December 2020 (late recovery)	Children	Overall vaccination, BCG, Hep B, Penta 1, OPV 1, PCV 1, Rota 1	Declines in vaccination for all vaccines during interruption, and declines and increases in vaccination during initial and late recovery depending on vaccine	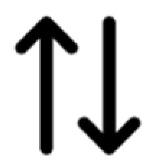
Shapira 2021 [35]	Democratic Republic of the Congo, Liberia, Malawi, Mali, Sierra Leone, Somalia	January 2018–February 2020 (baseline) compared to March–July 2020 (pandemic)	Children	BCG, Penta 3	Significant declines in vaccination for Penta 3 in three countries, significant declines in vaccination for BCG in two countries, significant increase in BCG in one country, and non-significant results for vaccinations in other countries	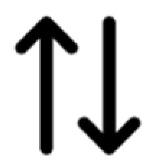
Wanyana 2021 [33]	Rwanda	March–April 2019 (baseline) compared to March–April 2020 (pandemic)	Children	BCG, Pol 0–3, IPV, DTP Hep B Hib 1–3, Pneumococcal 1–3, Rota 1–2, Measles + Rubella	Significant declines in 11 vaccines, non-significant declines in four vaccines, and non-significant increase in one vaccine	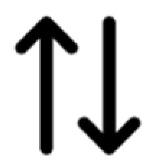

Note: Significance was determined using results displayed in Table 3. * Downward pointing arrows signify declines in vaccination. Downward and upward pointing arrows signify declines and increases in vaccination.

**Table 3 microorganisms-12-00573-t003:** Vaccination change results of included studies.

Citation	Measure Type	Results
Abid 2022 [31]	Daily vaccination coverage percent change	Overall vaccination coverage: −21.4% (*p* < 0.001) * BCG: −19% (*p* < 0.001) * DTP 2: −22% (*p* < 0.001) *, 3: −23% (*p* < 0.001) * Hep B: −6% (*p* < 0.001) * IPV: −23% (*p* < 0.001) * MCV: −28% (*p* < 0.001) * OPV 0: −18% (*p* < 0.001) *, 1: −19% (*p* < 0.001) *, 2: −22% (*p* < 0.001) *, 3: −23% (*p* < 0.001) *, 4: −28% (*p* < 0.001) * PCV 1: −21% (*p* < 0.001) *, 2: −23% (*p* < 0.001) *, 3: −26% (*p* < 0.001) * Penta 1: −19% (*p* < 0.001) * Rota 1: −20% (*p* < 0.001) *, 2: −23% (*p* < 0.001) *
Arsenault 2022 [39]	Vaccination coverage percent change	Fully vaccinated by 1 year: −0.91% (95% CI: −3.91, 2.1) BCG: −5.12% (95% CI: −14.73, 4.5) MCV: −1.69% (95% CI: −6.24, 2.87) Penta: −4.02% (95% CI: −10.84, 2.83) Pneumococcal: −3.97% (95% CI: −11.19, 3.26) Rota: −2.15% (95% CI: −7.73, 3.43)
Buonsenso 2020 [32]	Vaccination coverage percent change	BCG: −52.7% (*p* < 0.0005) * OPV 0: −52.7% (*p* < 0.0005) *, 1: −70.7% (*p* < 0.0005) *, 2: −78.9% (*p* < 0.0005) *, 3: −77.6% (*p* < 0.0005) * Penta 1: −70.7% (*p* < 0.0005) *, 2: −78.9% (*p* < 0.0005) *, 3: −77.6% (*p* < 0.0005) * PCV 1: −70.7% (*p* < 0.0005) *, 2: −78.9% (*p* < 0.0005) *, 3: −77.6% (*p* < 0.0005) * Rota 1: −70.7% (*p* < 0.0005) *, 2: −78.9% (*p* < 0.0005) * IPTI 1: −69.4% (*p* < 0.0005) *, 2: −65.9% (*p* < 0.0005) *, 3: −51.1% (*p* < 0.0005) * IPV: −77.6% (*p* < 0.0005) * MCV 1: −65.6% (*p* < 0.0005) *, 2: −83.7% (*p* < 0.0005) * Yellow fever: −65.6% (*p* < 0.0005) *
Burt 2021 [38]	Vaccination clinic attendance	Baseline: 5871 (95% CI: 5643, 6094) Pandemic: 906 (95% CI: 771, 2248) (*p* = 0.04) *
Connolly 2022 [30]	Vaccination coverage percent change	*Whole period* Overall vaccination coverage: 8% (95% CI: −12.1, 15.9) for Liberia, −2% (95% CI: −14.1, 12.6) for Malawi *Early pandemic* Overall vaccination coverage: −17% (95% CI: −39.1, −8) * for Liberia, 13.7% (95% CI: 2.4, 33.6) * for Malawi *Late pandemic* Overall vaccination coverage: 27% (95% CI: −12.5, 56.5) for Liberia, −9% (95% CI: −18, −3.9) * for Malawi
das Neves Martins Pires 2021 [41]	Vaccination coverage percent change	Overall vaccination coverage: −20% (*p* = 0.197) Complete vaccination: −18% (*p* = 0.544)
Gebreegziabher 2022 [34]	Vaccination coverage percent change	Fully vaccinated: −0.6% (*p* = 0.95) Penta 1: −0.3% (*p* = 0.94), 3: −4.7% (*p* = 0.27) MCV: 1.7% (*p* = 0.86)
Kassie 2021 [29]	Vaccination coverage proportion change	Penta: −0.033 (*p* = 0.011) * MCV: −0.031 (*p* = 0.008) *
Kotiso 2022 [40]	Vaccination coverage change	*February* Penta 3: −3.46 (95% CI: −6.63, −0.29) * *March* Penta 3: 0.02 (95% CI: −4.32, 4.37) *April* Penta 3: −6.09 (95% CI: −10.36, −1.82) * *May* Penta 3: −24.47 (95% CI: −30.56, −18.38) * *June* Penta 3: −15.31 (95% CI: −20.18, −10.45) *
Masresha 2020 [37]	Mean monthly vaccination dose percent change	Central African Republic DPT 3: −3%, MCV 1: −3% Chad DPT 3: 6%, MCV 1: 13% Democratic Republic of the Congo DPT 3: 1%, MCV 1: 2% South Sudan DPT 3: −7%, MCV 1: 9% Burundi DPT 3: −12%, MCV 1: −20% Eritrea DPT 3: −9%, MCV 1: 2% Rwanda DPT 3: −2%, MCV 1: −4%
Osei 2022 [36]	Vaccination coverage percent change	*Interruption* Vaccine administration: −38.3% BCG: −47.2% Hep B: −46.9% Penta 1: −43.1% OPV 1: −83.6% PCV 1: −42.4% Rota 1: −43.4% *Initial recovery* Vaccine administration: −15.1% BCG: −20% Hep B: −20% Penta 1: −33% OPV 1: −34% PCV 1: −33% Rota 1: −34% MCV: 79% Yellow fever: 88.9% *Late recovery* Vaccine administration: 1.9% BCG: 3% Hep B: 2.5% OPV 1: 22.7% PCV 1: −2% Rota 1: −2.6%
Shapira 2021 [35]	Vaccination coverage percent change	Democratic Republic of the Congo BCG: −1.4% (95% CI: −3.4, 0.6), Penta 3: −0.1% (95% CI: −1.2, 1) Liberia BCG: 0.3% (95% CI: −5.2, 5.8), Penta 3: −7.8% (95% CI: −13, −2.5) * Malawi BCG: 5.6% (95% CI: 0.5, 10.8) *, Penta 3: 1.2% (95% CI: −2.5, 4.9) Mali BCG: −11.8% (95% CI: −15.4, −8.2) *, Penta 3: −17.4% (95% CI: −22.6, −12.3) * Sierra Leone BCG: −7.4% (95% CI: −11.9, −2.9) *, Penta 3: −12.6% (95% CI: −19.1, −6.1) * Somalia BCG: −2.4% (95% CI: −8.9, 4.2), Penta 3: −3.6% (95% CI: −9.8, 2.6)
Wanyana 2021 [33]	Vaccination utilization rate change	BCG: −0.104 (*p* = 0.002) * Pol 0: −0.106 (*p* = 0.001) *, 1: −0.080 (*p* = 0.008) *, 2: −0.075 (*p* = 0.008) *, 3: −0.050 (*p* = 0.081) IPV: −0.047 (0.101) DTP HepB Hib 1: −0.080 (0.007) *, 2: −0.076 (0.007) *, 3: −0.050 (0.078) Pneumococcal 1: −0.081 (0.007) *, 2: −0.076 (0.007) *, 3: −0.050 (0.079) Rota 1: −0.083 (0.006) *, 2: −0.075 (0.009) * Measles + rubella 1: 0.014 (0.642), 2: −0.011 (0.625)

Note: Estimates are listed by vaccine dose when applicable. Doses range from 0 to 4 depending on the recommended schedule for each vaccine. * Significant results, *p* < 0.05.

**Table 4 microorganisms-12-00573-t004:** Summary of factors that influenced changes in routine childhood vaccination during the COVID-19 pandemic.

Themes	Factors	Studies
Communication challenges (n = 3)	Reduced communication for routine health services	Wanyana 2021 [33]
	Risk communication increased reluctancy to leave homes and get care	Abid 2022 [31]
	Social media misinformation	Shapira 2021 [35]
Fear of COVID-19 (n = 9)	Fear of contracting COVID-19 or transmission of COVID-19 in health facilities	Arsenault 2022 [30], Buonsenso 2020 [32], Burt 2021 [38], Connolly 2022 [30], das Neves Martins Pires 2021 [41], Gebreegziabher 2022 [34], Kotiso 2022 [40], Osei 2022 [36], Wanyana 2021 [33]
Financial barriers (n = 2)	Financial barriers to health services	Gebreegziabher 2022 [34]
	Inability to pay for healthcare due to loss of employment	Arsenault 2022 [39]
Health system practices (n = 5)	Weaker health systems were vulnerable to disruptions	Masresha 2020 [37]
	Resilience and structure of health systems Inadequate immediate care practices	Shapira 2021 [35] Kassie 2021 [29]
	Suspension of routine care practices for patients with COVID-19	Arsenault 2022 [39]
	Wait times increased	das Neves Martins Pires 2021 [41]
Lockdowns and emergency measures (n = 4)	Strict or lengthy lockdowns	Arsenault 2022 [39], Masresha 2020 [37]
	State of emergency raised alarms even with no lockdowns	Osei 2022 [36]
	Closure of borders	Abid 2022 [31]
Pre-existing factors (n = 2)	Pre-existing poor quality of services, poor road conditions, and lack of infrastructure for production of medical supplies	Wanyana 2021 [33]
	Prior experiences with epidemics and restrictions	Shapira 2021 [35]
Supply shortages (n = 4)	Vaccines out of stock	Connolly 2022 [30], Osei 2022 [36]
	Shortages in vaccines and other supplies, and disruptions in supply chains	Abid 2022 [31]
	Inadequate supply of PPE	Gebreegziabher 2022 [34]
Transportation barriers (n = 5)	Transportation restrictions and restricted movement	Connolly 2022 [30], Osei 2022 [36], Wanyana 2021 [33]
	Closure of public transport and increased price of transport	Burt 2021 [38]
	Motorcycle ambulances not used as often	das Neves Martins Pires 2021 [41]
Workforce changes (n = 3)	Redirected workforce to COVID-19 care	Gebreegziabher 2022 [34]
	Decrease in health professionals	Burt 2021 [38], das Neves Martins Pires 2021 [41]

## Data Availability

Data are contained within the articles and Appendix A.

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
