# Peer review of "Indirect Effects of the COVID-19 Pandemic on Routine Childhood Vaccination in Low-Income Countries: A Systematic Review to Set the Scope for Future Pandemics"

_microorganisms, 2024, doi:10.3390/microorganisms12030573_

Round 1

Reviewer 1 Report

Comments and Suggestions for Authors

1. Please add a comment about adults vaccination as well. 

2. Please comment on the WHO data on this issue (published vaccination rates globally, per country and vaccine etc)  

3. Please add the following article in your reference list: 

Maltezou et al. Decreasing routine vaccination rates in children in the COVID-19 era. Vaccine 2022;40:2525-2527.  

Author Response

Thank you for your feedback and constructive comments that have been valuable in enhancing the quality of this manuscript. We have provided responses to your comments below.

Reviewer 1

Comment 1: Please add a comment about adult vaccination as well. 

Response 1: Thank you for bringing up this point. We have added a comment on how adult vaccination was also impacted.

Comment 2: Please comment on the WHO data on this issue (published vaccination rates globally, per country and vaccine etc)  

Response 2: Thank you for this suggestion. We included global DTP vaccine coverage and coverage ranges using data from MMWR. These coverage rates match those reported by WHO and use the WHO global regions.  Since data are not available for all vaccines using the WHO WUENIC Trends tool, we chose to use DTP as an indicator for vaccination. We also chose to report by WHO global region instead of by country because WHO estimates can differ from government estimates and should be used with caution.

Comment 3: Please add the following article in your reference list: 

Maltezou et al. Decreasing routine vaccination rates in children in the COVID-19 era. Vaccine 2022;40:2525-2527.  

Response 3: Thank you for this comment. The article has been included in the reference list as reference 7.

Reviewer 2 Report

Comments and Suggestions for Authors

1) In the first section, the authors did not address properly the factors that may cause a reduction in the vaccination coverage during COVID-19 pandemic. Also the causes of vaccination coverage for other diseases is very superficial.

2) The methodology description was very good. 

3) In figure 1, the authors excluded 68 articles, Which were the reasons?

4) In figure 2, about the countries that were excluded, how different were social, economical and in health services than the countries includes?

5) In table 2, What was the threshold to considere a significant or non-significant reduction? Please make more clear that criteria. I can imagine that this is related with table 3, but that may be not obvious for all the readers. 

It was a nice systematic-review.

Author Response

Thank you for your feedback and constructive comments that have been valuable in enhancing the quality of this manuscript. We have provided responses to your comments below.

Reviewer 2

Comment 1: In the first section, the authors did not address properly the factors that may cause a reduction in the vaccination coverage during COVID-19 pandemic. Also the causes of vaccination coverage for other diseases is very superficial.

Response 1: Thank you for this important point. We have enhanced the introduction to include factors that may cause a reduction in vaccination coverage.

Comment 2: The methodology description was very good. 

Response 2: Thank you for this kind comment. We have not made any changes to the methodology section.

Comment 3: In figure 1, the authors excluded 68 articles, Which were the reasons?

Response 3: Thank you for bringing up this point. We have added the reasons for article exclusion during title and abstract screening. We are not able to give exact numbers for the exclusion reasons like during full text screening because the systematic review software we used did not collect that information.

Comment 4: In figure 2, about the countries that were excluded, how different were social, economical and in health services than the countries includes?

Response 4: Thank you for this comment. We addressed any differences between included and not included countries in the discussion section of the paper.

Comment 5: In table 2, What was the threshold to consider a significant or non-significant reduction? Please make more clear that criteria. I can imagine that this is related with table 3, but that may be not obvious for all the readers. 

Response 5: Thank you for this comment. We added a note to the bottom of table 2 to make significance more clear.

Comment 6: It was a nice systematic-review.

Response 6: We sincerely appreciate this comment on the quality of our review.